# Fundamental Limits of Transfer Learning in Binary Classifications

## Abstract

A critical performance barrier in modern machine learning is scarcity of labeled data required for training state of the art massive models, especially in quickly emerging problems with lack of extensive data sets or scenarios where data collection and labeling is expensive/time consuming. Transfer learning is gaining traction as a promising technique to alleviate this barrier by utilizing the data of a related but different *source* task to compensate for the lack of data in a *target* task where there are few labeled training data. While there has been many recent algorithmic advances in this domain, a fundamental understanding of when and how much one can transfer knowledge from a related domain to reduce the amount of labeled training data is far from understood. We provide a precise answer to this question for binary classification problems by deriving a novel lower bound on the generalization error that can be achieved by *any* transfer learning algorithm (regardless of its computational complexity) as a function of the amount of source and target samples. Our lower bound depends on a natural notion of distance that can be easily computed on real world data sets. Other key features of our lower bound are that it applies to any arbitrary source/target data distributions and requires minimal assumptions that enables it application to a broad range of problems. We also consider a more general setting where there are more than one source domains for knowledge transfer to the target task and develop new bounds on generalization error in this setting. We also corroborate our theoretical findings on real image classification and action recognition data sets. These experiments demonstrate that our natural notion of distance is indicative of the difficulty of knowledge transfer between different pairs of source/target tasks, allowing us to investigate the effect of different sources on the target generalization error. Furthermore, to evaluate the sharpness of our bounds we compare our developed lower bounds with upper-bounds achieved by transfer learning base-lines that utilize weighted empirical risk minimization on the combination of source(s) and target data sets.

## 1 Introduction

Modern machine learning models such as deep neural networks have enjoyed wide success in many domains Krizhevsky et al. (2012). The success of such deep models critically relies on an enormous amount of data required for training these massive models. For instance, GPT3 which is the state of the art model for natural language process has 175 billion parameters and requires a data set of size 45 terabytes for training. However, in new or emerging application domains it is often extremely difficult or costly to gather such large labeled training data.

A promising approach to this problem has been via *transfer learning* which aims at leveraging abundant available labeled data from a related *source* task to reduce the amount of labeled data required for the *target* task Pan & Yang (2009); Weiss et al. (2016). From a practical perspective transfer learning has been rather successful empirically. In particular, state of the art transfer learning approaches based on pretrained models and fine tuning has led to significant improvements on various benchmark datasets. Despite this empirical success however there is a huge gap between theory and practice in transfer learning and the fundamental limits and benefits of transfer learning are not well understood. Key challenging questions include: What is an appropriate notion of similarity between different tasks and how can it be quantitatively defined and computed on real data? What is the best

achievable accuracy of any transfer learning algorithm with only a limited number of source and target samples? How does this accuracy depend on the number of samples and the similarity between the source and target tasks?

While the answer to these challenging questions are still not fully understood, they have indeed attracted a lot of interesting theoretical work in this area Galanti et al. (2016). We will discuss this literature in thorough detail in Section 2. In this paper, we take a step towards answering the aforementioned key questions enabling a better understanding of the fundamental limits of transfer learning. We focus on binary classifications where the goal is to learn a classifier from a hypothesis class with a finite VC-dimension. This covers most contemporary classification models including the training deep neural networks for binary classification. In this setting, we first define a natural notion of similarity between source and target tasks via the performance of the best source hypothesis on the target task. Then equipped with this notion of similarity, we derive a statistical minimax lower bound on the target generalization error in terms of the number of labeled data from source and target tasks as well as the VC dimension of the hypothesis class and the similarity between source and target tasks. Furthermore, we extend this result to the case where there are multiple sources with different similarity to the target. Our results demonstrate that sources with high similarity to the target are more effective at reducing the target generalization error. Towards bridging the theory-practice gap in transfer learning we also demonstrate the utility of our theoretical result in concrete applications. Indeed, a key feature of our result is that our lower bounds can be easily and efficiently computed on real data sets and apply to a broad class of practical settings.

In summary our key contributions are as follows:

- We develop a novel statistical minimax lower bound on the generalization error that can be achieved by any transfer learning algorithm as a function of the amount of source and target samples and a natural notion of similarity between source and target tasks.

- A key features of our lower bound (including our notion of similarity) is that it can be easily computed on real world data sets. Furthermore, our lower bound holds for any source/target distribution and applies with minimal assumptions to a wide variety of contemporary learning models including deep neural networks.

- We investigate the sharpness of our lower bounds and demonstrate their utility via experiments on action recognition and image classification.

## 2 PRIOR WORKS

A closely related literature to transfer learning is domain adaptation where there is no or very few labeled target data and the goal is to adapt the hypothesis learned on the source domain to achieve a low target generalization error Chen et al. (2019); Blitzer et al. (2007); Azizzadenesheli et al. (2018); Long et al. (2016); Shen et al. (2018). Most of this literature assume that source and target share a common labeling rule but there is a shift in the marginal distributions. There are many upper bounds for the target generalization error in this setting this setting. For instance, Ben-David et al. (2007; 2010) gives an upper bound for the target generalization error in terms of source generalization error and a divergence measure between the domains that can be estimated by finitely many unlabeled data from the source and target. In another work Mansour et al. (2009) introduces a new discrepancy distance and generalizes the results of Ben-David et al. (2007) for a wide family of loss functions using Rademacher complexity. Similar to this setting, but for multiple source domain adaption scheme, Mansour et al. (2021) proposes a family of algorithms based on the idea of model selection under the assumption that target distribution is close to some convex combination of sources. A more recent work Lei et al. (2021) studies linear regression under shift distribution including covariate shift (i.e. conditional distributions of source and target are the same) as well as model shift (i.e. only distributions of the features of the source and target are the same) and develops algorithms achieving near optimal minimax risk in this setting.

In addition to upper bounds, there are also a few results which provide lower bounds for target generalization error. David et al. (2010) provides impossibility results under the assumption of covariate shift and small discrepancy of unlabeled distributions. Mousavi Kalan et al. (2020) studies transfer learning with one hidden layer neural networks for regression problems. This result defines a notion of similarity between the source and target tasks based on a distance between the ground truth

parameters of the source and target networks. Using this distance this paper develops a statistical minimax lower bound for the target generalization error in terms of the number of source and target samples as well as the defined similarity of the source and target under the distribution shift with the assumption that the features are generated by Gaussian distributions. Compared to Mousavi Kalan et al. (2020) our result has quite a few unique advantages: (1) We do not assume that the source and target data are generated according to a planted (teacher) network and our results now even hold in the agnostic setting. (2) Mousavi Kalan et al. (2020) applies to regression problems but this result covers classification (3) Mousavi Kalan et al. (2020) only considered one-hidden layer neural networks for predicting the labels of extracted features. In this result we can handle arbitrary deep neural networks. (4) Our notion of similarity between the source and target distributions can be much more easily estimated by using only a few target data without the need for estimating the ground truth target parameters which requires lots of labeled target data.

More closely related to this work Hanneke & Kpotufe (2019) derives a minimax lower bound for target generalization error in binary classification under the assumption of a relaxed version of covariate shift and small transfer exponent parameter which is defined to measure the discrepancy of the source and target distributions. Our work differs from this previous work as except for assuming the VC dimension of the model is finite we do not make any further assumptions. This makes our results applicable in a much broader set of classifications or decision making problems. Furthermore, our lower bound can be evaluated on real data sets and serve as a guideline to practitioners helping them decide when utilizing additional knowledge from a source domain is useful for a given target task.

Most of the literature in transfer learning try to provide sufficiency and necessity results by deriving upper and lower bounds for target generalization error in a relatively general setting. However, these papers often require a variety of assumptions to find the optimal classifier in a target domain in closed form. For instance, Karbalayghareh et al. (2019; 2018) defines a joint prior distribution of source and target domains using a Wishart distribution which relate the source and target tasks and then makes it possible to study and understand the transferability between domains. Furthermore, in this setting, the authors develop a closed form optimal Bayesian transfer learning and demonstrate its advantage over a classifier obtained by only target data. Related to this setting but for regressions, Karbalayghareh et al. (2018) obtains the optimal Bayesian transfer learning under setting of joint Gaussian feature/label distribution. In contrast with the above in our paper we do not make any assumptions about the distribution of the data.

## 3    PROBLEM FORMULATION

We consider a transfer learning problem where there are some labeled training data from a source task and a target task with the goal of inferring a hypothesis function with small generalization error in the target task. More specifically, we assume have $n_S$ and $n_T$ source and target labeled data where each training data consists of an input/feature as well as an output/label. We denote the source and training data by $(\boldsymbol{x}_S, y_S) \sim \mathbb{P}$ and $(\boldsymbol{x}_T, y_T) \sim \mathbb{Q}$, respectively, where $y_S, y_T \in \{0, 1\}$ and $\mathbb{P}, \mathbb{Q}$ are the joint feature-label distributions of source and target data. Additionally, we assume that source and target features/inputs share a same domain, $\boldsymbol{x}_S, \boldsymbol{x}_T \in \chi$, and $\mathcal{H} \subset 2^\chi$ denotes a fixed hypothesis class with $d_\mathcal{H}$ VC-dimension.

In transfer learning the goal is to find a hypothesis from $\mathcal{H}$ that minimizing the *target excess risk* defined below based on a combination of source and target data.

**Definition 1** *(Excess risk) For a hypothesis function $h \in \mathcal{H}$ and source and target label-feature data generated according to distributions $\mathbb{P}$ and $\mathbb{Q}$ ($(\boldsymbol{x}_S, y_S) \sim \mathbb{P}$ and $(\boldsymbol{x}_T, y_T) \sim \mathbb{Q}$), we define the source and target excess risks as follows*

$$\mathcal{E}_T(h) = \mathbb{Q}[h(\boldsymbol{x}_T) \neq y_T] - \mathbb{Q}[h_T^*(\boldsymbol{x}_T) \neq y_T]$$

*and*

$$\mathcal{E}_S(h) = \mathbb{P}[h(\boldsymbol{x}_S) \neq y_S] - \mathbb{P}[h_S^*(\boldsymbol{x}_S) \neq y_S]$$

*where $h_T^* = \arg\min_{h \in \mathcal{H}} \mathbb{Q}[h(\boldsymbol{x}_T) \neq y_T]$ and $h_S^* = \arg\min_{h \in \mathcal{H}} \mathbb{P}[h(\boldsymbol{x}_S) \neq y_S]$*

Next, we need to define an appropriate notion of distance between the source and target. In the literature of domain adaptation, where the conditional expectation remains unchanged and there is

only a shift in input distributions, it is common to define the distance as the error of performance of the best source hypothesis in the target task. We also define the distance between source and target as the target excess risk of the best source hypothesis.

**Definition 2** *(Transfer distance) We define the transfer distance between a source and a target with distributions $\mathbb{P}$ and $\mathbb{Q}$ as follows*

$$\rho(\mathbb{P}, \mathbb{Q}) := \mathbb{Q}[h_S^*(\boldsymbol{x}_T) \neq y_T] - \mathbb{Q}[h_T^*(\boldsymbol{x}_T) \neq y_T] \tag{3.1}$$

Since we aim to derive a minimax lower bound for transfer learning in binary classifications, we consider the class of pairs of distributions whose transfer distance is within a fixed number $\Delta$. As we will elaborate further in Remark 7 below this notion of distance can be easily estimated/computed in practice.

## 4 MAIN RESULTS

In this section we characterize the fundamental limits of transfer learning in binary classifications by deriving a minimax lower bound via information-theoretic arguments.

**Theorem 1** *Consider a transfer learning problem where there are $n_S$ and $n_T$ number of source as well as target data and the hypothesis class $\mathcal{H}$ has VC dimension $d_{\mathcal{H}}$ obeying $d_{\mathcal{H}} \geq 10$. Furthermore, suppose that $\hat{h} = \hat{h}(S_{\mathbb{P}}, S_{\mathbb{Q}})$ is an estimated hypothesis for the target task using source and target data in which $S_{\mathbb{P}}$ and $S_{\mathbb{Q}}$ denote i.i.d. feature-label data pairs $\{(\boldsymbol{x}_S^{(i)}, y_S^{(i)})\}_{i=1}^{n_S}$ and $\{(\boldsymbol{x}_T^{(i)}, y_T^{(i)})\}_{i=1}^{n_T}$ generated according to the source and target distributions $\mathbb{P}$ and $\mathbb{Q}$. Fix a transfer distance $\Delta < 0.99$. Then for any $\hat{h}$ there exists $(\mathbb{P}, \mathbb{Q})$ with $\rho(\mathbb{P}, \mathbb{Q}) \leq \Delta$ and a universal constant $c$ such that*

$$\underset{S_{\mathbb{P}}, S_{\mathbb{Q}}}{Prob}\left(\mathcal{E}_T(\hat{h}) > c \cdot \epsilon(n_S, n_T, d_{\mathcal{H}}, \Delta)\right) \geq \frac{3 - 2\sqrt{2}}{8},$$

*where*

$$\epsilon(n_S, n_T, d_{\mathcal{H}}, \Delta) = \sqrt{\frac{1}{\frac{n_T}{d_{\mathcal{H}}} + \frac{n_S}{d_{\mathcal{H}} + n_S \Delta}}}.$$

*This also implies that*

$$\inf_{\hat{h}} \sup_{\rho(\mathbb{P}, \mathbb{Q}) \leq \Delta} \underset{S_{\mathbb{P}}, S_{\mathbb{Q}}}{\mathbb{E}}\left[\mathcal{E}_T(\hat{h})\right] \geq c \cdot \epsilon(n_S, n_T, d_{\mathcal{H}}, \Delta). \tag{4.1}$$

**Remark 1** *The bound above characterizes the fundamental limits of transfer learning by providing a lower bound on the excess risk of any algorithm (regardless of computational tractability) as a function of the number of source and target training data, the similarity/distance between the source and target tasks and the dimension of the hypothesis class used.*

**Remark 2** *The assumption $\Delta < 0.99$ in the statement of Theorem 1 is just made for simplifying the analysis and the upper bound of $0.99$ can be replaced by any constant in the interval $(0, 1)$.*

**Remark 3** *One can show that the numerical constant $c$ in equation 4.1 obeys $c > \frac{3 - 2\sqrt{2}}{48}$.*

**Remark 4** *(Connection to PAC learning) We note that the well-known agnostic PAC learning result for a single task gives a lower bound of $c \cdot \sqrt{\frac{d_{\mathcal{H}}}{n}}$ where $n$ is the number of samples of the task. Theorem 1 recovers this result when there is not any source task, namely $n_S = 0$, and the transfer learning problem reduces to learning a task without any prior knowledge from the source.*

**Remark 5** *(Identical source and target) When the source and target tasks are identical, then the transfer learning problem reduces to learning a single task with $n_S + n_T$ training data. Theorem 1, also leads to the same conclusion in this special case as when the source and target data are identical $\Delta = 0$ and thus $\epsilon = \sqrt{\frac{d_{\mathcal{H}}}{n_S + n_T}}$ which states that the lower bound is proportional to reciprocal of combination of source and target samples as expected.*

**Remark 6** *(Sharpness in a special case) We note that the above lower bound is known to be tight in special cases. For instance when there is a small amount of source data and $\Delta$ is rather large, the lower bound reduces to $\sqrt{\frac{d_{\mathcal{H}}}{n_T}}$ which is known to be tight based on known agnostic PAC learning bounds.*

**Remark 7** *(How to apply Theorem 1 in practical settings.) In this remark we explain how Theorem 1 can be applied when using contemporary machine learning models involving artificial neural networks. In this case, the hypothesis class corresponds to all neural networks with a fixed architecture but different parameters. It is known that the class of neural networks with a fixed architecture has finite VC dimension and Harvey et al. (2017) gives upper and lower bounds for VC dimension of neural networks with ReLU activation functions. Thus, to apply Theorem 1, one only needs to have an estimate of the transfer distance per Definition 2. We note that the transfer distance 3.1 consists of two terms: To estimate the first term, we note that $h_S^*$ can be easily estimated due to the abundance of source data in most applications. Also with an estimate of $h_S^*$ in hand one can estimate $\mathbb{Q}[h_S^*(\boldsymbol{x}_T) \neq y_T]$ rather accurately using a simple empirical average with a few target test data as well-known concentration of bounded functions imply that this empirical average is well concentrated around $\mathbb{Q}[h_S^*(\boldsymbol{x}_T) \neq y_T]$. Up on first glance it seems that estimating the second term which corresponds to the lowest possible error in the target domain among the hypothesis class, requires a large amount of labeled target data which is not available in a practical problem. However, in an overparametrized setting, it is typical to assume that there exists a network which achieves very small target generalization error so we can ignore the second term in most practical problems. Finally we note that as stated earlier the lower bound on the target excess risk gives an estimate of what generalization performance we can expect with a certain number of source and target samples. Furthermore, by comparing the estimated transfer distance of different pairs of tasks, we can find the pairs that are more suitable for transfer learning. This knowledge can in turn significantly reduce the required number of target samples to achieve a certain accuracy.*

Next, we extend our result to a multiple source transfer learning setup where instead of only one source task there are several source tasks available and the goal is to transfer knowledge from multiple sources to a given target task to achieve a small target generalization error.

**Theorem 2** *Suppose that there are $n_{S_1}, n_{S_2}, ..., n_{S_N}$ number of samples from $N$ source tasks as well as $n_T$ number of samples from a target task and the hypothesis class $\mathcal{H}$ has VC dimension $d_{\mathcal{H}}$ obeying $d_{\mathcal{H}} \geq \max(N+9, N/2)$. Furthermore, suppose that $\hat{h} = \hat{h}(S_{\mathbb{P}_1}, S_{\mathbb{P}_2}, ..., S_{\mathbb{P}_N}, S_{\mathbb{Q}})$ is an estimated hypothesis for the target task using $N$ sources and target data where $S_{\mathbb{P}_j}$ and $S_{\mathbb{Q}}$ denote i.i.d. data $\{(\boldsymbol{x}_{S_j}^{(i)}, y_{S_j}^{(i)})\}_{i=1}^{n_{S_j}}$ and $\{(\boldsymbol{x}_T^{(i)}, y_T^{(i)})\}_{i=1}^{n_T}$ generated according to souce and target distributions $\mathbb{P}_j$ and $\mathbb{Q}$ for $j = 1, ..., N$. Fix transfer distances $\{\Delta_j\}_{j=1}^N$ where $0 \leq \Delta_j \leq 1$. Then for any $\hat{h}$ there exists $(\mathbb{P}_1, ..., \mathbb{P}_M, \mathbb{Q})$ with $\rho(\mathbb{P}_j, \mathbb{Q}) \leq \Delta_j$ and a universal constant $c$ such that*

$$\underset{S_{\mathbb{P}_1}, ..., S_{\mathbb{P}_N}, S_{\mathbb{Q}}}{Prob}\left(\mathcal{E}_T(\hat{h}) > c \cdot \epsilon(n_{S_1}, ..., n_{S_N}, n_T, d_{\mathcal{H}}, \Delta_1, ..., \Delta_N)\right) \geq \frac{3 - 2\sqrt{2}}{8},$$

*where*

$$\epsilon(n_{S_1}, ..., n_{S_N}, n_T, d_{\mathcal{H}}, \Delta_1, ..., \Delta_N) = \sqrt{\frac{1}{\frac{n_T}{d_{\mathcal{H}}} + \frac{n_{S_1}}{d_{\mathcal{H}} + n_{S_1}\Delta_1} + ... + \frac{n_{S_N}}{d_{\mathcal{H}} + n_{S_N}\Delta_N}}}.$$

*This in turn implies that*

$$\inf_{\hat{h}} \sup_{\substack{\rho(\mathbb{P}_j, \mathbb{Q}) \leq \Delta_j \\ j=1,...N}} \underset{S_{\mathbb{P}_1}, ..., \mathbb{P}_N, S_{\mathbb{Q}}}{\mathbb{E}}\left[\mathcal{E}_T(\hat{h})\right] \geq c \cdot \epsilon(n_{S_1}, ..., n_{S_N}, n_T, d_{\mathcal{H}}, \Delta_1, ..., \Delta_N). \tag{4.2}$$

**Remark 8** *Similar to the previous theorem, Theorem 2 provides a minimax lower bound for target excess risk with the key distinction that now it applies in the setting where there are multiple source with different transfer distances to the target. This theorem characterizes the exces risk achievable by any algorithm as a function of these transfer distances as well as the number of samples from the different sources and the target data. Theorem 2 indicates that the more sources we have, the better performance we can achieve in the target domain. However, this performance gain maybe marginal*

*for source tasks that have a large transfer distance to the target or where there are very few training data. In these cases of course it may be more computationally efficient to discard these sources given the marginal improvement in the generalization performance suggested by this theorem.*

**Remark 9** *(Identical sources) if all the source tasks are identical, then there are effectively $n_{S_1} + ... + n_{S_N}$ number of source samples and by Theorem 1 the lower bound would be $\sqrt{\frac{1}{\frac{n_T}{d_{\mathcal{H}}} + \frac{\sum_{j=1}^{N} n_{S_j}}{d_{\mathcal{H}} + \Delta \sum_{j=1}^{N} n_{S_j}}}}$. Theorem 2 also gives the same order wise lower bound as*

$$\sqrt{\frac{1}{\frac{n_T}{d_{\mathcal{H}}} + \sum_{j=1}^{N} \frac{n_j}{d_{\mathcal{H}} + \Delta n_j}}} \leq \sqrt{\frac{1}{\frac{n_T}{d_{\mathcal{H}}} + \frac{\sum_{j=1}^{N} n_{S_j}}{d_{\mathcal{H}} + \Delta \sum_{j=1}^{N} n_{S_j}}}} \leq \sqrt{N} \cdot \sqrt{\frac{1}{\frac{n_T}{d_{\mathcal{H}}} + \sum_{j=1}^{N} \frac{n_j}{d_{\mathcal{H}} + \Delta n_j}}}$$

**Remark 10** *(Infinitely many source samples) When $\Delta_i > 0$ and $n_{S_i} \to \infty$, the fraction $\frac{n_{S_i}}{d_{\mathcal{H}} + n_{S_i} \Delta_i}$ saturates at $\frac{1}{\Delta_i}$ which shows that when the source and target have positive distance, the source can never compensate for the target samples.*

**Remark 11** *In the lower bound, the product terms $\Delta_i n_{S_i}$ appear which indicate that a source with large transfer distance can sometimes be as useful as a source with small transfer distance when there is a large amount of training data available from that source.*

## 5 EXPERIMENTAL RESULTS

In this section we evaluate our theoretical results on real data sets for action recognition and image classification tasks. By estimating the parameters appearing in Theorem 1 for different pairs of tasks, we first plot the lower bounds and then by running weighted empirical risk minimization investigate the sharpness of the bounds. We also investigate the effectiveness of different source tasks with different transfer distances on the target generalization error.

### 5.1 ACTION RECOGNITION

**Experimental setup.** We first perform experiments on the UCF101 action recognition data set. We pick CricketBowling and TableTennis videos from UCF101 as the target task as well as three different pairs of classes as the source tasks: 1- CricketBowling and BaseballPitch, 2- Cricketshot and Archery, 3- BasketballDunk and Basketball. We pass the videos through an i3d network pretrained on kinetics400 Carreira & Zisserman (2017) with the fully connected top classifier removed and extract the corresponding features of dimension 2048 from the raw videos. We then work with the extracted features instead of the raw videos.

**Training.** We train a one hidden layer neural network with 15 number of hidden units and ReLU activation functions for each pair of data sets. Table 3 consists of test accuracy on CricketBowling vs. TableTennis, when using the network trained on each source task. We use these accuracies for deriving the corresponding lower bounds. Furthermore, we run weighted empirical risk minimization as a simple transfer learning approach to find some upper bounds on the target generalization error. Given $n_S$ and $n_T$ number of source and target samples, for estimating the corresponding one hidden layer neural network parameters we minimize the following weighted empirical risk

$$\min_{\boldsymbol{W}_1, \boldsymbol{W}_2} \frac{1 - \lambda}{n_T} \sum_{i=1}^{n_T} \text{Cost}(\boldsymbol{W}_2 \text{ReLU}(\boldsymbol{W}_1 \boldsymbol{x}_T^{(i)}), y_T^{(i)}) + \frac{\lambda}{n_S} \sum_{i=1}^{n_S} \text{Cost}(\boldsymbol{W}_2 \text{ReLU}(\boldsymbol{W}_1 \boldsymbol{x}_S^{(i)}), y_S^{(i)}) \quad (5.1)$$

where the function Cost denotes the logistic regression cost and $\lambda \in \{0, 0.2, 0.4, 0.6, 0.8, 1\}$. We then pick the lambda which minimizes the target test error.

**Results.** First we calculate the transfer distance by Definition 2 for each source/target pairs using Table 1. To this end, we assume that best target generalization error is zero and using the Table 1 we obtain the transfer distance for each pair which is demonstrated in Table 2. As it can be observed by Table 2, the pair of Source1 and Target has the lowest transfer distance among other pairs since both of the source and target tasks share a same class which is CricketBowling. Furthermore, Table 2 determines which pairs are more suitable for transferring the source knowledge to the target.

| Task | Test accuracy of Target using the source network |
|---|---|
| Target: CricketBowling vs. TableTennis | - |
| Source1: CricketBowling vs. Baseball Pitch | 0.946 |
| Source2: Cricketshot vs. Archery | 0.61 |
| Source3: BasketballDunk vs. Basketball | 0.52 |

Table 1

| pair of tasks | $\rho$(Source, Target) |
|---|---|
| (Source1, Target) | 0.053 |
| (Source2, Target) | 0.39 |
| (Source3, Target) | 0.48 |

Table 2: Transfer distance of pairs of source and target on UCF101 action recognition.

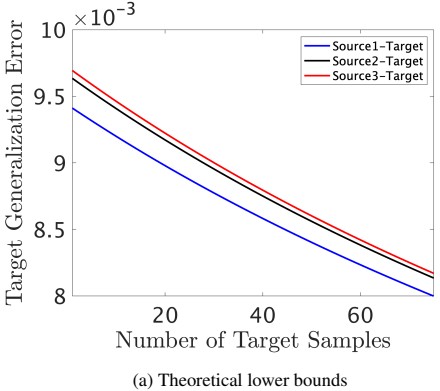

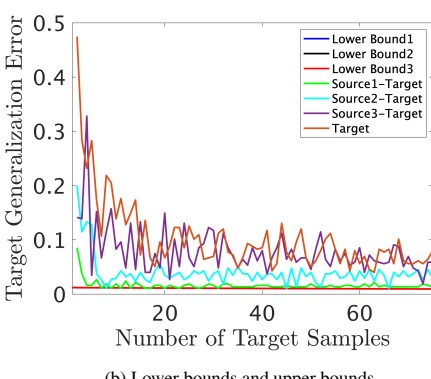

(a) Theoretical lower bounds         (b) Lower bounds and upper bounds

Figure 1: (a) depicts our lower bounds for three pairs of source and target tasks on action classification. (b) depicts the lower bounds along with the upper bounds obtained via weighted empirical risk minimization.

Next, we draw the lower bound curves for each pair in Fig 1a. To this end, we need to find the VC dimension of the hypothesis class which consists of neural networks with the architecture of $2048 * 15 * 1$ with ReLU activation functions. Theorem 1 in Harvey et al. (2017) gives a lower bound for VC dimension of neural networks with ReLU activation functions by $\frac{1}{640} W L \log_2 \frac{W}{L}$ where $W$ and $L$ are the number of parameters and layers, respectively. Then in Figure 1b we plot the lower bounds along with the upper bounds obtained via Formula 5.1 for three different pairs of source and target as well as using only target samples. We obtained these upper bounds by running Formula 5.1 five times and then averaging the results. Fig 1b shows that when the distance of a source from the target is small it would be more effective in achieving small target generalization error. We would like to mention that in all of these plots we choose the same number of source samples for each pair.

Figure 2 shows the average $\lambda$, the weight appearing in Formula 5.1, when the number of target samples is 100 to 150. It shows that in the pair Source1 and Target the average $\lambda$ is high which demonstrate the usefulness of the source in the target task. Furthermore, the small value of $\lambda$ in the pair Source3 and Target suggests that when the transfer distance is high, source samples are no longer usefull.

## 5.2 IMAGE CLASSIFICATION

**Experimental setup.** In this section we focus on image classification tasks and utilize Theorem 1 to recognize appropriate pairs of tasks that are suitable for transfer learning. We choose some classes of the DomainNet data set Peng et al. (2019) as source and target tasks. We pick Clock and Ambulance from DomainNet Clipart for the target task and three different pairs of classes as the source tasks: 1- Clock and Ambulance, 2- Cricketshot and TableTennis, 3- TableTennis and FrontCraw. Here we

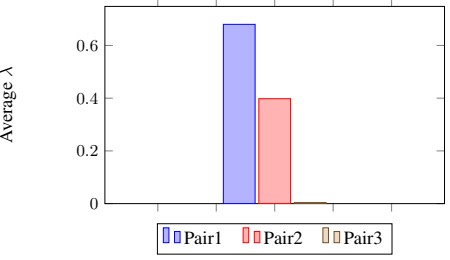

Figure 2: Average $\lambda$ in weighted empirical risk minimization for three different pairs of source and target tasks for action recognition.

| Task | Test Accuracy of Target using the source network |
|---|---|
| Target: Clock vs. Ambulance (Clipart) | - |
| Source1: Clock vs. Ambulance (Sketch) | 0.916 |
| Source2: Clock vs. Crow (Sketch) | 0.697 |
| Source3: Crow vs. Basket (Sketch) | 0.65 |

Table 3

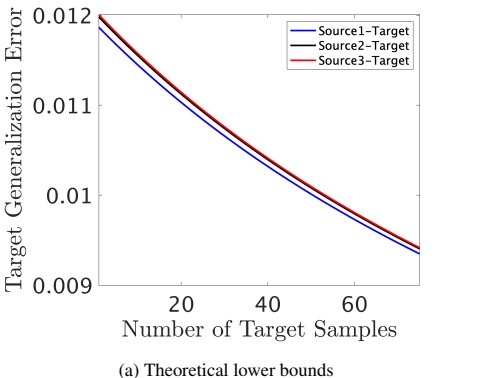

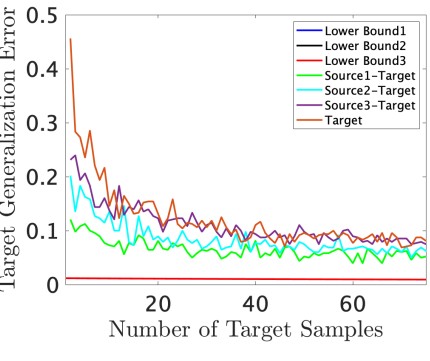

(a) Theoretical lower bounds

(b) Lower bounds and upper bounds

Figure 3: (a) depicts our lower bounds for three pairs of source and target tasks on image classification. (b) depicts the lower bounds along with the upper bounds obtained via weighted empirical risk minimization.

| pair of tasks | $\rho(\text{Source}, \text{Target})$ |
|---|---|
| (Source1, Target) | 0.083 |
| (Source2, Target) | 0.3 |
| (Source3, Target) | 0.35 |

Table 4: Transfer distance of pairs of source and target on DomainNet image classifications].

use ResNet50 network pretrained on Imagenet for extracting features of dimension 2048 and in the sequel we work with the extracted features rather than the raw image data.

**Training.** We train a one hidden layer neural network with 15 number of hidden units and ReLU activation functions for each of pairs of the tasks. Table 3 includes the test accuracy on the target task when using the networks trained on different sources, which is necessary for estimating/calculating the transfer distance as demonstrated in Table 4. Similar to the subsection 5.1, we also run weighted empirical risk minimization for finding upper bounds for the pairs of the source and target tasks.

**Results.** Similar to the previous section on action recognition, using Table 3 we can obtain the transfer distances and based on this distance we can identify suitable pairs of source and target tasks

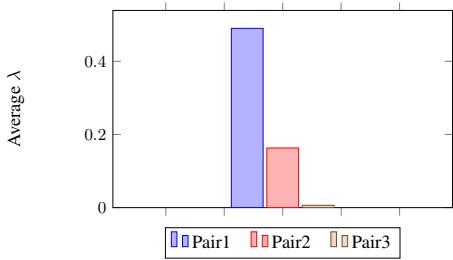

Figure 4: Average $\lambda$ in weighted empirical risk minimization for three different pairs of source and target tasks for image classification.

for transfer learning. In the pair1 Source and target tasks share the same objects which are Clock and Ambulance which results in low transfer distance. In pair2, still one of the objects which is Clock is the same in the source and target and we can see that the transfer distance for pair2 is lower than that for pair3. Then we plot the lower bounds in Fig 3a and the corresponding upper bounds obtained by weighted empirical risk minimization in Fig 3b. One can see that sources that are closer to the target according to our notion of distance are more effective in achieving small target generalization error.

CricketBowling is common both in the source and Target1. This suggests that these tasks are similar to each other and the estimated transfer distance conforms with this intuition. Furthermore, CricketBowling and Cricketshot are intuitively similar to one another and this is also reflected in the lower transfer distance between source and Target2.

In Fig 4 we plot the average $\lambda$, the weight appearing in Formula 5.1 when the number of target samples varies from 150 to 200. 4 demonstrates that when a source is close to the target the weight of source risk in weighted empirical risk becomes high which shows the effectiveness of source samples in achieving small target generalization error.

## 6 PROOF OUTLINE

The main idea of proof is based on the following proposition proved in Tsybakov (2009)

**Proposition 1** *[Theorem 2.5 of Tsybakov (2009)] Assume that $M \geq 2$ and the function $d(\cdot, \cdot)$ is a semi-distance. Also suppose that $\{P_{\theta_j}\}_{\theta_j \in \Theta}$ is a family of distributions indexed over a parameter space, $\Theta$, and $\Theta$ contains elements $\theta_0, \theta_1, ..., \theta_M$ such that:*

*(i) $d(\theta_i, \theta_j) \geq 2s > 0, \ \ \forall \, 0 \leq j < k \leq M$*

*(ii) $P_j \ll P_0, \ \ \forall \, j = 1, ..., M,,$ and*

$$\frac{1}{M} \sum_{j=1}^{M} \mathcal{D}_{kl}(P_j | P_0) \leq \alpha \log M$$

*with $0 < \alpha < 1/8$ and $P_j = P_{\theta_j}, j = 0, 1, ..., M$ and $\mathcal{D}_{kl}$ denotes the KL-divergence. Then*

$$\inf_{\hat{\theta}} \sup_{\theta \in \Theta} P_\theta(d(\hat{\theta}, \theta) \geq s) \geq \frac{\sqrt{M}}{1 + \sqrt{M}} \left( 1 - 2\alpha - \sqrt{\frac{2\alpha}{\log M}} \right)$$

Based on Proposition 1 we construct a family of pairs of distributions, namely source and target distributions, whose transfer distances satisfy the $\Delta$-constraint. To do so we pick some points from the domain $\chi$ shattered by the hypothesis class and define appropriate distributions on this set of points. Furthermore, this family of distributions are indexed in the space of $\{-1, 1\}^d$ which can be a metric space using Hamming distance. In order to satisfy the condition (i) in Proposition 1, the indexes have to be well separated which can be achieved using the well-known Gilbert-Varshamov's bound. Finally we show that estimating a parameter with small hamming distance is equivalent to estimating an appropriate hypothesis with small excess risk error.

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

# 7 APPENDIX

## 7.1 PROOF OF THEOREM 1

We also use the following famous result in information theory known as Gilbert-Varhsamov's bound for packing argument.

**Proposition 2** *(Lemma 2.9 of Tsybakov (2009)) Let $d \geq 8$. Then there exists a subset $\{w^{(0)}, ..., w^{(M)}\}$ of $\Omega = \{-1, 1\}^d$ such that $w^{(0)} = (1, 1, ..., 1)$,*

$$dist(w^{(j)}, w^{(k)}) \geq \frac{d}{8}, \ \forall \, 0 \leq j < k \leq M \text{ and } M \geq 2^{d/8},$$

*where $dist(w, w') = \sum_{k=1}^{d} I(w_k \neq w'_k)$ is the Hamming distance between binary sequences $w = (w_1, ..., w_d)$ and $w' = (w'_1, ..., w'_d)$.*

We will also use the following lemma proved in Hanneke & Kpotufe (2019). We would like to mention that some ideas of the proof are similar to those in Hanneke & Kpotufe (2019). However, as discussed in section 2, the problem setting of Hanneke & Kpotufe (2019) is different from that of this work which results in constructing a different set of distributions.

**Lemma 1** *Let $0 < \epsilon < 1/2$ and $z \in \{-1, 1\}$. Then*

$$\mathcal{D}_{kl}\bigg( Ber\big(1/2 + (z/2) \cdot \epsilon\big), Ber\big(1/2 - (z/2) \cdot \epsilon\big) \bigg) \leq c_0 \cdot \epsilon^2 \text{ for some } c_0 \leq 4 \text{ independent of } \epsilon$$

Now we are in place to provide the proof of Theorem 1. Let $d = d_{\mathcal{H}} - 2$ and pick $\boldsymbol{x}_{-1}, \boldsymbol{x}_0, ..., \boldsymbol{x}_d$ from $\chi$ shattered by $\mathcal{H}$.

Next, we construct a family of pairs of distributions $(\mathbb{P}_w, \mathbb{Q}_w)$ indexed by $w \in \{-1, 1\}^d$ where $\{-1, 1\}^d$ is the parameter space playing the role of $\Theta$ in Proposition 1. For the following, fix $\epsilon = c_1 \cdot \epsilon(n_S, n_T, d_{\mathcal{H}}, \Delta) \leq \frac{1}{2}$ for some constant $c_1$ to be determined later in proof and $\epsilon(n_S, n_T, d_{\mathcal{H}}, \Delta)$ is defined in Theorem 1.

**Distribution $\mathbb{Q}_w$:** $\mathbb{Q}_w$ is composed of a marginal and a conditional distribution, namely $\mathbb{Q}_w = \mathbb{Q}_{\boldsymbol{x}}^w \times \mathbb{Q}_{y|\boldsymbol{x}}^w$. We define the marginaldistributions as follows:

$$\mathbb{Q}_{\boldsymbol{x}}^w(\boldsymbol{x} = \boldsymbol{x}_{-1}) = \Delta$$
$$\mathbb{Q}_{\boldsymbol{x}}^w(\boldsymbol{x} = \boldsymbol{x}_0) = 0.99 - \Delta$$
$$\mathbb{Q}_{\boldsymbol{x}}^w(\boldsymbol{x} = \boldsymbol{x}_i) = \frac{1}{100d} \ \text{ for } \ i = 1, .., d$$

For the conditional distributions:

$$\mathbb{Q}_{y|\boldsymbol{x}}^w(y = 1|\boldsymbol{x} = \boldsymbol{x}_{-1}) = \mathbb{Q}_{y|\boldsymbol{x}}^w(y = 1|\boldsymbol{x} = \boldsymbol{x}_0) = 1$$
$$\mathbb{Q}_{y|\boldsymbol{x}}^w(y = 1|\boldsymbol{x} = \boldsymbol{x}_i) = 1/2 + (w_i)\epsilon \ \text{ for } \ i = 1, .., d$$

**Distribution $\mathbb{P}_w$**: $\mathbb{P}_w$ is composed of a marginal and a conditional distribution, namely $\mathbb{P}_w = \mathbb{P}_{\boldsymbol{x}}^w \times \mathbb{P}_{y|\boldsymbol{x}}^w$. We define the marginal distributions as follows:

$$\mathbb{P}_{\boldsymbol{x}}^w(\boldsymbol{x} = \boldsymbol{x}_{-1}) = \mathbb{P}_{\boldsymbol{x}}^w(\boldsymbol{x} = \boldsymbol{x}_0) = 1/2\big(1 - \frac{d}{d + n_S\Delta}\big)$$

$$\mathbb{P}_{\boldsymbol{x}}^w(\boldsymbol{x} = \boldsymbol{x}_i) = \frac{1}{d + n_S\Delta} \quad \text{for } i = 1, .., d$$

For the conditional distributions:

$$\mathbb{P}_{y|\boldsymbol{x}}^w(y = 1|\boldsymbol{x} = \boldsymbol{x}_{-1}) = 0$$
$$\mathbb{P}_{y|\boldsymbol{x}}^w(y = 1|\boldsymbol{x} = \boldsymbol{x}_0) = 1$$
$$\mathbb{P}_{y|\boldsymbol{x}}^w(y = 1|\boldsymbol{x} = \boldsymbol{x}_i) = 1/2 + (w_i)\epsilon \text{ for } i = 1, .., d$$

**Verifying $\rho(\mathbb{P}_w, \mathbb{Q}_w) \leq \Delta$**: Bayes classifier of the domain generated by $\mathbb{P}_w$ is as follows:

$$h_S^*(\boldsymbol{x}_{-1}) = 0$$
$$h_S^*(\boldsymbol{x}_0) = 1$$
$$h_S^*(\boldsymbol{x}_i) = 1 \text{ if } w_i = 1 \text{ , otherwise } h_S^*(\boldsymbol{x}_i) = 0 \text{ for } i = 1, .., d$$

Similarly for the domain generated by $\mathbb{Q}_w$, we have

$$h_T^*(\boldsymbol{x}_{-1}) = h_T^*(\boldsymbol{x}_0) = 1$$
$$h_T^*(\boldsymbol{x}_i) = 1 \text{ if } w_i = 1 \text{ , otherwise } h_T^*(\boldsymbol{x}_i) = 0 \text{ for } i = 1, .., d$$

So $h_S^*$ and $h_T^*$ disagree only on $\boldsymbol{x}_{-1}$ which implies that

$$\rho(\mathbb{P}_w, \mathbb{Q}_w) = \mathbb{Q}[h_S^*(\boldsymbol{x}_T) \neq y_T] - \mathbb{Q}[h_T^*(\boldsymbol{x}_T) \neq y_T] = \Delta$$

Since we want to derive a lower bound for the minimax risk stated in Theorem 1, among the hypotheses that they agree on $\boldsymbol{x}_i$ for $i = 1, ..., d$, the hypothesis that outputs $\boldsymbol{x}_{-1}$ and $\boldsymbol{x}_0$ as 1 results in a smaller target error. Hence, we can restrict ourselves to $\tilde{\mathcal{H}}$ which is the projection of $\mathcal{H}$ onto $\{-1, 1\}^d$ with the constraint that $h(\boldsymbol{x}_{-1}) = h(\boldsymbol{x}_0) = 1$ for all $h \in \tilde{\mathcal{H}}$. Furthermor, for any $w, w' \in \{-1, 1\}^d$ we have

$$\mathcal{E}_T(h_{w'}) = \frac{\text{dist}(w, w')}{100d} \cdot \epsilon, \ \forall\, h_{w'} \in \tilde{\mathcal{H}}$$

when the target domain is generated by $\mathbb{Q}_w$

**Reduction to a packing**: By using Proposition 2, we can get a subset $\Sigma$ of $\{-1, 1\}^d$ whose cardinality is $M \geq 2^{d/8}$ and for any $w, w'$ belonging to $\Sigma$ we have $\text{dist}(w, w') \geq d/8$. Furthermore, for any $w, w' \in \Sigma$ we have

$$\mathcal{E}_T(h_{w'}) \geq \frac{d}{8} \cdot \frac{\epsilon}{100d} = \frac{\epsilon}{800}$$

On the other hand, there is a bijective map between $\{-1, 1\}^d$ and elements of $\tilde{\mathcal{H}}$ and any classifier $\hat{h} : \{\boldsymbol{x}_i\} \to \{0, 1\}$ with $\hat{h}(\boldsymbol{x}_{-1}) = \hat{h}(\boldsymbol{x}_0) = 1$ can be reduced to a $w \in \{-1, 1\}^d$. So we can choose $\Sigma$ as the set of indices in Proposition 1 with Hamming distance as the semi-metric and the expression $P_w(\text{dist}(\hat{w}, w) > d/8)$ translates into $P_w(\mathcal{E}_T(h_{\hat{w}}) > c \cdot \epsilon)$.

**KL divergence bound (part (ii) of Proposition 1)**: Define $P_w = \mathbb{P}_w^{n_S} \times \mathbb{Q}_w^{n_T}$. For any $w, w' \in \Sigma$ we have

$$\mathcal{D}_{kl}(P_w|P_{w'}) = n_S \cdot \mathcal{D}_{kl}(\mathbb{P}_w|\mathbb{P}_w') + n_T \cdot \mathcal{D}_{kl}(\mathbb{Q}_w|\mathbb{Q}_{w'})$$
$$= n_S \cdot \underset{\mathbb{P}_{\boldsymbol{x}}}{\mathbb{E}} \mathcal{D}_{kl}(\mathbb{P}_{y|\boldsymbol{x}}^w|\mathbb{P}_{y|\boldsymbol{x}}^{w'}) + n_T \cdot \underset{\mathbb{Q}_{\boldsymbol{x}}}{\mathbb{E}} \mathcal{D}_{kl}(\mathbb{Q}_{y|\boldsymbol{x}}^w|\mathbb{Q}_{y|\boldsymbol{x}}^{w'})$$
$$= n_S \cdot \sum_{i=1}^{d} \frac{1}{d + n_S\Delta} \mathcal{D}_{kl}(\mathbb{P}_{y|\boldsymbol{x}_i}^w|\mathbb{P}_{y|\boldsymbol{x}_i}^{w'}) + n_T \cdot \sum_{i=1}^{d} \frac{1}{100d} \mathcal{D}_{kl}(\mathbb{Q}_{y|\boldsymbol{x}_i}^w|\mathbb{Q}_{y|\boldsymbol{x}_i}^{w'})$$
$$\leq n_S \cdot \frac{d}{d + n_S\Delta} c_0\epsilon^2 + n_T \cdot \frac{1}{100} c_0\epsilon^2$$
$$\leq c_0 c_1^2 \cdot d$$

if $c_1 < \frac{1}{6}$ then $c_0 c_1^2 < \frac{1}{8}$ and we can apply Proposition 1.

## 7.2 PROOF OF THEOREM 2

Proof of Theorem 2 is similar to that of Theorem 1. However, we construct different target and source probability distributions.

Let $d = d_{\mathcal{H}} - N - 1$ and pick $x_{-M}, ..., x_0, x_1, ..., x_d$ from $\chi$ shattered by $\mathcal{H}$. Then we construct a family of distributions $(\mathbb{P}_w^{(1)}, ..., \mathbb{P}_w^{(N)}, \mathbb{Q}_w)$ indexed by $w \in \{-1, 1\}^d$. Let $\epsilon = c_1 \cdot \epsilon(n_{S_1}, ..., n_{S_N}, n_T, d_{\mathcal{H}}, \Delta_1, ..., \Delta_N)$ for some constant $c_1 < 1$ to be determined later in proof. Furthermore, without loss of generality assume that $1 \geq \Delta_1 \geq \Delta_2 \geq ... \geq \Delta_N \geq 0$.

**Distribution $\mathbb{Q}_w$:** $\mathbb{Q}_w$ is composed of a marginal and a conditional distribution, namely $\mathbb{Q}_w = \mathbb{Q}_{\boldsymbol{x}}^w \times \mathbb{Q}_{y|\boldsymbol{x}}^w$. We define the marginal distributions as follows:

$$\mathbb{Q}_{\boldsymbol{x}}^w(\boldsymbol{x} = \boldsymbol{x}_{-i}) = \Delta_i - \Delta_{i+1} \text{ for } i = 1, ..., N-1 \text{ and } \mathbb{Q}_{\boldsymbol{x}}^w(\boldsymbol{x} = \boldsymbol{x}_{-N}) = \Delta_N$$
$$\mathbb{Q}_{\boldsymbol{x}}^w(\boldsymbol{x} = \boldsymbol{x}_0) = 0.99 - \Delta_1$$
$$\mathbb{Q}_{\boldsymbol{x}}^w(\boldsymbol{x} = \boldsymbol{x}_i) = \frac{1}{100d} \text{ for } i = 1, .., d$$

For the conditional distributions:

$$\mathbb{Q}_{y|\boldsymbol{x}}^w(y = 1 | \boldsymbol{x} = \boldsymbol{x}_{-i}) = 1 \text{ for } i = 1, ..., N$$
$$\mathbb{Q}_{y|\boldsymbol{x}}^w(y = 1 | \boldsymbol{x} = \boldsymbol{x}_0) = 1$$
$$\mathbb{Q}_{y|\boldsymbol{x}}^w(y = 1 | \boldsymbol{x} = \boldsymbol{x}_i) = 1/2 + (w_i)\epsilon \text{ for } i = 1, ..., d$$

**Distribution $\mathbb{P}_w^{(i)}$:** $\mathbb{P}_w^{(i)}$ is composed of a marginal and a conditional distribution, namely $\mathbb{P}_w^{(i)} = \mathbb{P}_{\boldsymbol{x}}^{w\,(i)} \times \mathbb{P}_{y|\boldsymbol{x}}^{w\,(i)}$. We define the marginal distributions as follows:

$$\mathbb{P}_{\boldsymbol{x}}^{w\,(i)}(\boldsymbol{x} = \boldsymbol{x}_{-j}) = \frac{1}{N+1}\left(1 - \frac{d}{d + n_{S_i}\Delta_i}\right) \text{ for } j = 1, ..., N$$
$$\mathbb{P}_{\boldsymbol{x}}^{w\,(i)}(\boldsymbol{x} = \boldsymbol{x}_0) = \frac{1}{N+1}\left(1 - \frac{d}{d + n_{S_i}\Delta_i}\right)$$
$$\mathbb{P}_{\boldsymbol{x}}^{w\,(i)}(\boldsymbol{x} = \boldsymbol{x}_j) = \frac{1}{d + n_{S_i}\Delta_i} \text{ for } j = 1, .., d$$

For the conditional distributions:

$$\mathbb{P}_{y|\boldsymbol{x}}^w(y = 1 | \boldsymbol{x} = \boldsymbol{x}_{-j}) = 0 \text{ if } j \geq i, \text{ otherwise } \mathbb{P}_{y|\boldsymbol{x}}^w(y = 1 | \boldsymbol{x} = \boldsymbol{x}_{-j}) = 1 \text{ for } j = 1, ..., N$$
$$\mathbb{P}_{y|\boldsymbol{x}}^w(y = 1 | \boldsymbol{x} = \boldsymbol{x}_0) = 1$$
$$\mathbb{P}_{y|\boldsymbol{x}}^w(y = 1 | \boldsymbol{x} = \boldsymbol{x}_j) = 1/2 + (w_j)\epsilon \text{ for } j = 1, .., d$$

**Verifying $\rho(\mathbb{P}_w^{(i)}, \mathbb{Q}_w) \leq \Delta_i$:**

Bayes classifier of the domain generated by $\mathbb{P}_w^{(i)}$ is as follows:

$$h_{S_i}^*(\boldsymbol{x}_{-j}) = 0 \text{ if } j \geq i, \text{ otherwise } h_{S_i}^*(\boldsymbol{x}_{-j}) = 1 \text{ for } j = 1, ..., N$$
$$h_{S_i}^*(\boldsymbol{x}_0) = 1$$
$$h_{S_i}^*(\boldsymbol{x}_j) = 1 \text{ if } w_j = 1, \text{ otherwise } h_{S_i}^*(\boldsymbol{x}_j) = 0 \text{ for } j = 1, .., d$$

Similarly for the domain generated by $\mathbb{Q}_w$, we have

$$h_T^*(\boldsymbol{x}_{-j}) = 1 \text{ for } j = 1, ..., N$$
$$h_T^*(\boldsymbol{x}_0) = 1$$
$$h_T^*(\boldsymbol{x}_j) = 1 \text{ if } w_j = 1, \text{ otherwise } h_T^*(\boldsymbol{x}_j) = 0 \text{ for } j = 1, .., d$$

So $h_{S_i}^*$ and $h_T^*$ disagree on $\boldsymbol{x}_{-i}, .., \boldsymbol{x}_{-N}$ which implies that

$$\rho(\mathbb{P}_w^{(i)}, \mathbb{Q}_w) = \mathbb{Q}[h_{S_i}^*(\boldsymbol{x}_T) \neq y_T] - \mathbb{Q}[h_T^*(\boldsymbol{x}_T) \neq y_T] = \Delta_i$$

With the same argument we used in the proof of Theorem 1 we can restrict ourselves to $\tilde{\mathcal{H}}$ which is the projection of $\mathcal{H}$ with the constraint that $h(\boldsymbol{x}_{-N}) = ... = h(\boldsymbol{x}_{-1}) = h(\boldsymbol{x}_0) = 1$ for all $h \in \tilde{\mathcal{H}}$.

The rest of the proof is exactly the same except the part regarding the KL divergence bound.

**KL divergence bound:** Define $P_w = \mathbb{P}_w^{(1)^{n_{S_1}}} \times ... \times \mathbb{P}_w^{(N)^{n_{S_N}}} \times \mathbb{Q}_w^{n_T}$.

$$
\begin{aligned}
\mathcal{D}_{kl}(P_w|P_{w'}) &= \sum_{j=1}^{N} n_{S_j} \cdot \mathcal{D}_{kl}(\mathbb{P}_w^{(j)}|\mathbb{P}_{w'}^{(j)}) + n_T \cdot \mathcal{D}_{kl}(\mathbb{Q}_w|\mathbb{Q}_{w'}) \\
&= \sum_{j=1}^{N} n_{S_j} \cdot \underset{\mathbb{P}_{\boldsymbol{x}}^{(j)}}{\mathbb{E}}\, \mathcal{D}_{kl}(\mathbb{P}_{y|\boldsymbol{x}}^{w}{}^{(j)}|\mathbb{P}_{y|\boldsymbol{x}}^{w'}{}^{(j)}) + n_T \cdot \underset{\mathbb{Q}_{\boldsymbol{x}}}{\mathbb{E}}\, \mathcal{D}_{kl}(\mathbb{Q}_{y|\boldsymbol{x}}^{w}|\mathbb{Q}_{y|\boldsymbol{x}}^{w'}) \\
&= \sum_{j=1}^{N} n_{S_j} \cdot \sum_{i=1}^{d} \frac{1}{d + n_{S_j}\Delta_j}\mathcal{D}_{kl}(\mathbb{P}_{y|\boldsymbol{x}_i}^{w}{}^{(j)}|\mathbb{P}_{y|\boldsymbol{x}_i}^{w'}{}^{(j)}) + n_T \cdot \sum_{i=1}^{d} \frac{1}{100d}\mathcal{D}_{kl}(\mathbb{Q}_{y|\boldsymbol{x}_i}^{w}|\mathbb{Q}_{y|\boldsymbol{x}_i}^{w'}) \\
&\leq \sum_{j=1}^{N} n_{S_j} \cdot \frac{d}{d + n_{S_j}\Delta_j}c_0\epsilon^2 + n_T \cdot \frac{1}{100}c_0\epsilon^2 \\
&\leq c_0 c_1^2 \cdot d
\end{aligned}
$$

for small enough $c_1$ we can apply Proposition 1.

### 7.3 ADDITIONAL EXPERIMENTAL RESULTS

In section 5 we fix number of source samples and vary the number of target samples. Here in order to investigate the effect of source samples on the target generalization error, we fix the number of target samples at $n_T = 3$ and vary the number of source samples. Fig 5 depicts the theoretical lower bounds along with the upper bounds obtained by empirical risk minimization for image classifications. We use the same source/target pairs as used in section 5.2. Fig 5 demonstrates that Source1 is more helpful in reducing the target generalization error because it has a low distance from the target. Furthermore, it shows that increasing the number of source samples is useful up to a point and beyond that point the error saturates and does not decrease further as discussed in Remark 10.

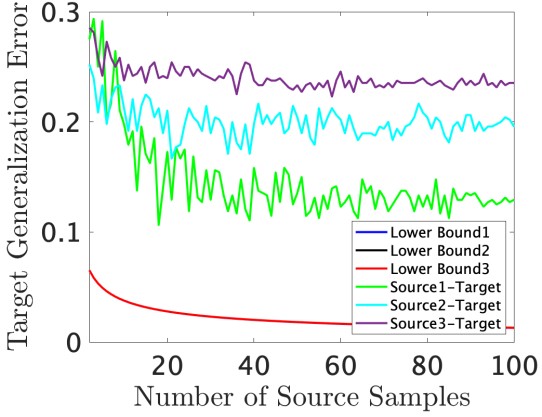

Figure 5: Depicts the lower bounds along with the upper bounds obtained via weighted empirical risk minimization. In this setting the number of target samples is fixed at $n_T = 3$

