# OpenReview forum: "Fundamental Limits of Transfer Learning in Binary Classifications"
_ICLR.cc/2022/Conference — ICLR 2022 Submitted_

### Official Review · Reviewer_dGvD · 2021-11-02

**Correctness:** 3
**Technical Novelty And Significance:** 2
**Empirical Novelty And Significance:** 3
**Recommendation:** 6
**Confidence:** 2

**Main Review:**

It gives new bounds with minimal assumptions, but
1.	It is better to give more comparison between the new bound and the existing ones.
2.	It is only for binary classes, but in real transfer learning, it is usually multi-class learning problems.
3.     In experiments, it only shows that the bound is affected by the number of source and target training data, the similarity between source and target tasks. Moreover, how to find if the bound is tight or not?



**Summary Of The Paper:**

This paper derives a novel lower bound on the generalization error of transfer learning as a function of the amount of source and target samples. It does not depend on the source/target data distributions and requires minimal assumptions. It also derive a bound for multi-source domains.

**Summary Of The Review:**

Based on the above review, the paper is slightly above the acceptance bound.

---

> ### Author Response · Authors · 2021-11-22
> **Response to reviewer dGvD**
>
> We thank the reviewer for a thorough reading of the manuscript and helpful comments.
>
> 1. There are mainly two major lines of works in the literature of transfer learning. One major line is related to finding transfer learning algorithms which provide upper bounds and sufficiency results. Another major line is related to finding lower bounds and necessity results, and our paper lies in this category. In the prior works section we have thoroughly discussed the previous results , and to the best of our knowledge those are the main bounds existing in the literature. However, if the reviewer has suggestions on other lines of works we would appreciate pointing us to those and we will happily discuss them.
>
> 2. Most of the prior works in transfer learning are also limited to binary classifications [1,2,3]. Finding a lower bound for the most general setting of transfer learning, even for binary classification, with minimal assumptions is already challenging and from the theoretical perspective, this work may also be useful for multi-class learning problem since the notion of transfer distance defined in this paper is general and can be also used for multi-class problems.
>
> Furthermore, we believe our results can be extended to the multiclass setting by using notion of "Natarajan dimension", which is an extension of VC dimension for multi-class settings, to adapt our lower bound for multi-class transfer learning. However, this is beyond the scope of this paper and requires fundamentally new results.
>
> 3. In the experiemnts, we have drawn both the lower bound and upper bounds obtained by weighted ERM algorithm. one can see that the gap between the them is negligible which shows the tightness of the lower bound empirically. That said, we plan to investigate their tightness theoretically in future work. But this requires fundamentally different techniques as it requires bound on performance of different algorithms and is hence beyond the scope of the current work.
>
> [1]: Steve Hanneke, Samory Kpotufe; On the Value of Target Data in Transfer Learning.
>
> [2]:Shai Ben-David, John Blitzer, Koby Crammer, Alex Kulesza, Fernando Pereira,
> Jennifer Wortman Vaughan; A theory of learning from different domains.
>
> [3]: H. Zhao, R. T. d. Combes, K. Zhang, and G. J. Gordon; On learning invariant representation for domain adaptation

---

### Official Review · Reviewer_pmYF · 2021-11-02

**Correctness:** 3
**Technical Novelty And Significance:** 3
**Empirical Novelty And Significance:** 2
**Recommendation:** 6
**Confidence:** 2

**Main Review:**

Transfer learning is a natural, well-studied approach to dealing with scenarios with limited data. The authors make two main contributions in this work: a new distance measure (called “transfer distance”) between learning problems, and a clean generalization lower bound for when transfer learning can be helpful. The empirical experiments give a convincing argument that transfer distance is a relevant quantity in practice, and deserves further study. Overall, the paper is very well written and should be of interest to a general learning audience.

On the other hand, the work is not without weaknesses. The main focus of the paper is on proving a lower bound, but there is no indication of whether the new transfer term in the bound is tight. While some empirical experiments are done using ERM transfer methods, the results do not seem particularly convincing in this aspect. Second, the authors make a major point of emphasizing that transfer distance is easy to estimate (see Remark 7). This is actually false, at least in the worst-case. Having a good estimate $h$ of $h^*_S$ over $S$ is not sufficient to estimate $Q[h_S^*(x_T) \neq y_T]$, because the marginal weight where $h$ and $h^*_S$ differ could shift dramatically between the source and target distributions. This needs to be clarified, and to recover the result it would be nice to see conditions on S and T where it remains true.


Minor comments for the authors:

1. Citations for empirical success in first paragraph?

2. typos: is differs, resullts,

3. Covariate shift is never defined.

4. On page 3 it is stated that domain adaptation requires the conditional to stay the same—should this say covariate shift?

**Summary Of The Paper:**

This work studies lower bounds on transfer learning in distribution-free classification, a technique for leveraging related source data S for some target classification problem T. This is a useful idea in practice since common ML algorithms such as deep learning require massive amounts of labeled data, which can often be prohibitively expensive. If we can leverage data from an existing problem and transfer the knowledge, we can get around this issue. The authors in this work provide a lower bound on the generalization error of transfer learning dependent on a natural measure of distance between the source problem S and target problem T (in particular, this is given by the error of the best hypothesis on S evaluated on T). They then provide empirical evidence that their distance measure captures the difficulty of transfer learning in practice, and that it can be estimated using available source and target data.

**Summary Of The Review:**

Overall, I recommend weak acceptance of this work: the authors introduce a natural notion of distance and give a clean lower bound, but the evidence of tightness is weak and the discussion of estimating transfer distance needs to be corrected.

EDIT: I have read the authors' response and my review remains largely unchanged. I believe with the addition of either a theoretical upper bound or more convincing empirical verification would lead to a very strong paper, but I can only recommend weak acceptance of its current incarnation. It could also be helpful to include experiments on estimating transfer distance in reasonable settings, since the assumption that $Q[h^*_S(x_T) \neq \hat{h}(x_T)]$ is small does potentially limit applicability to scenarios without any major shifts in the underlying distributional weights.

---

> ### Author Response · Authors · 2021-11-22
> **Response to reviewer pmYF**
>
> We thank the reviewer for a thorough reading of the manuscript and helpful comments.
>
> 1- With respect to the tightness of the lower bound, we note that deriving a lower bound without establishing tightness is already challenging, particularly since we derived the lower bound for transfer learning in the most general setting for the first time. As explained in remark 6, the lower bound recovers the PAC lower bound which is known to be sharp. Based on this fact we conjecture that the lower bound is sharp and it can be achieved by algotithms (perhaps even by modifications of weighted ERM). We have also empirically demonstrate the sharpness of the bound to some extent.
>
> About the second comment, first of all we have the following triangle inequality:
>
> $$
> Q[h_{S}^{*}(x_T)\neq y_T] \leq Q[h^{\*}_S(x_T)\neq \hat{h}(x_T)]+Q[\hat{h}(x_T)\neq y_T]
> $$
>
> We can estimate the second term of RHS by using a few test data from the target. Furthermore, we know that having abundat of source samples we can make sure that $P[h^{\*}_S(x_T)\neq \hat{h}(x_T)]$ is small enough. So the question is that if $P[h^{\*}_S(x_T)\neq \hat{h}(x_T)]$ is small enough can we conclude that $Q[h^{\*}_S(x_T)\neq \hat{h}(x_T)]$ is also small enough? For most of the practical settings the answer is yes. Because if source and target share the same support or the support of source contains the support of target and if we also assume the continuity of the distributions this implication holds. Essentially this holds if the two distributions are absolutely continuous with respect to each other. we will clarify this further.
>
>  Minor Comments:
>  1- Thanks for the comments. We have revised the paper and highlighted the modified parts.

---

### Official Review · Reviewer_TfQT · 2021-11-02

**Correctness:** 2
**Technical Novelty And Significance:** 2
**Empirical Novelty And Significance:** 2
**Recommendation:** 3
**Confidence:** 4

**Main Review:**

Strong points:

1. The paper is generally well organized. It is easy to understand the purpose and contribution of this paper.
2. The literature review is clear. The relation to prior work is explained in detail.

Weak points

1. (P4) The lower bound in (4.1) is not sharp. Consider a general case of transfer learning problem for bounds, i.e., $n_S>>n_T>>0$. In this case, the second term in the denominator is canceled, and the bound is reduced to $\sqrt{\frac{1}{n_T/d_H+1}}$. This means that a large size of source data cannot improve the bound, i.e., this lower bound is very loose for a transfer learning problem.

2. The theoretical contribution is not significant enough. The derivation of a lower bound based on VC dimension setup is not a novel idea.
3. (P3)(3.1） The distribution assumption that $\rho(P,Q)$ is smaller than a fixed number $\Delta$, is very strong. In addition, the lower bound in (4.1) depends on this $\Delta$. Therefore, the claim in the abstract, "the lower bound does not depend on the source/target data distributions" does not hold.
4. Experiments are running the datasets with a size of about 100. This is too small to confirm a theoretical bound. Generally speaking, the order data size needs to be large enough to avoid the effect of constant in the bound.

Suggestions:
1. Run the experiments on datasets with large sizes such as 100K to check the bounds.
2. Theoretical prove the lower bound is sharp, which would be a significant theoretical contribution.



**Summary Of The Paper:**

This paper provides a lower bound on the generalization error for binary classification problems. This bound can be achieved by any transfer learning algorithm (regardless of its computational complexity) as a function of the amount of source and target samples. In addition, this lower bound and can be generalized to the multi-source case.

**Summary Of The Review:**

The initial recommendation of the paper. is "reject, not good enough."

---

> ### Author Response · Authors · 2021-11-22
> **Response to reviewer TfQT**
>
> We thank the reviewer for helpful comments. Given some misunderstandings that might have arisen, as clear from our response below, we hope the reviewer reconsiders their score.
>
> 1. First, please note that the lower bound in this case would converge to $\sqrt{\frac{1}{\frac{n_T}{d}+\frac{1}{\Delta}}}$ not $\sqrt{\frac{1}{\frac{n_T}{d}+1}}$. Based on the comment number 3 of the reviewer, we think that  there is a misunderstanding here and the reviewer thinks that $\Delta$ is a constant number. However, it is a parameter varying between [0,1] which determines the set of source/target distributions over which the minimax risk is derived. In other words, when $\Delta$ is small, the minimax risk is derived over the source and target distributions whose distance is small, and when it is close to 1, the minimax risk is derived over the source and target distributions whose distance is large.
>
> We don't think your argument suggests that lower bound is not tight. When $n_S>>n_T$ and $\Delta>0$ (positive discrepancy) the lower bound saturates at ($\sqrt{\frac{1}{\frac{n_T}{d}+\frac{1}{\Delta}}}$) when $n_S$ goes to infinity, and increasing the number of source samples further is no longer helpful. Because when there is a positive discrepancy between source and target, source samples are useful up to a point and beyond that they cannot compensate for the target samples. Furthermore, [1] proves the sharpness of a recently derived minimax lower bound for transfer learning for some cases in which the lower bound saturates when $n_S$ goes to infinity. This indicates that the given example by the reviewer does not imply that the derived lower bound is loose.
>
> 2. Although transfer learning has been proven to be useful in practice, there are very few theoretical results on the limits of transfer learning, and the foundational understanding of “when” and “what” to transfer still remains elusive. There are some papers deriving lower bounds for transfer learning, but all of them make restrictive assumptions. [2] derives lower bounds only for one hidden layer neural networks with features whose distributions are Gaussian. [3] derives lower bounds under the assumption of a relaxed version of covariate shift.
>
> This paper, for the first time, proves fundamental limits of transfer learning in the most general setting with minimal assumptions that enables it application to a broad range of practical problems. Furthermore, This work makes a bridge between the theory of classical PAC learning and transfer learning.
>
> With regard to VC dimension setup,VC dimension is one of the fundamental quantities appearing in most of the theoretical results of machine learning. In the literature of transfer learning specifically, most of the important results are based on this fundamental notion and they have also been shown to be useful in practice [4]. We think the appearance of this important and somewhat natural notion, does not diminish the novelty of our results.
>
> 3. As previously mentioned, there is a misunderstanding here. $\Delta$ is not a fixed number rather a parameter varying between [0,1]. We derived a minimax lower bound over the class of distributions whose distance is less than $\Delta$ which could be any number between [0,1]. Maybe the expression "fix the transfer distance $\Delta$" in Theorem 1 was the source of this misunderstanding. When one derives a minimax lower bound, one first fixes the class of distributions that are considered and then derives the lower bound. But this does not mean $\Delta$ is a universal constant.
>
> With regard to the second part of the reviewer's comment, we agree with the reviewer that the mentioned expression can cause confusion and we need to clarify it. The lower bound actually depends on the distribution of the source and target. However, it can be applied to any class of distributions as we did not make any assumptions on the class of distributions of the source and target. We would like to mention that many prior work make some assumptions that the distributions belong to a particular class such as Gaussian [2]. We have revised the mentioned sentence in the paper to reflect this. Thanks!
>
> 4. First we would like to emphasize that our result is not asymptotic and holds for any number of samples. Furthermore, the constant is universal and does not depend on the number of samples. Additionally, since our result only applies to binary classifications, we were limited to pick samples from only two classes. However, to the best of our knowledge, all well-known datasets have order of 100 samples per class. But if the reviewer has any suggestions for another dataset with a larger number of samples per class, we would be happy to consider and add that.
>
> [1]:Near-Optimal Linear Regression under Distribution Shift,[2]:Minimax lower bounds for transfer learning with linear and one-hidden layer neural networks, [3]:On the Value of Target Data in Transfer Learning,[4]:A theory of learning from different domains.

---

> > ### Comment · Reviewer_TfQT · 2021-11-29
> > **Post Rebuttal**
> >
> > Dear authors,
> > Thanks for the detailed response. The following are my additional responses.
> >
> > 1. As stated in P3 (3.1) and the "sentence class of pairs of distributions whose transfer distance is within a fixed number." Once P and Q are determined in source and target data distribution, the \Delta is also determined. Therefore, $\Delta$ cannot be small enough and $1/\Delta$ cannot reach the order of $n_T$ in the denominator. Therefore, this means that a large size of source data cannot improve the bound.
> >
> > In addition, in the response Point 1. "Because when there is a positive discrepancy between source and target, source samples are useful up to a point and beyond that they cannot compensate for the target samples." This is an assumption on source/target data distributions, which contradicts with "it applies to any arbitrary source/target data distributions." in the updated abstract claims.
> >
> > 2. In the response Point 2: "This paper, for the first time, proves fundamental limits of transfer learning in the most general setting with minimal assumptions that enables it application to a broad range of practical problems."
> > In fact, there are many assumptions which are not "minimal" in this paper. For example, Theorem 1 assumes "VC dimension >10" which many VC dimension-related papers do not require.
> >
> > 3. Thanks for your updates in the edit. Point 3 claims, "However, it can be applied to any class of distributions as we did not make any assumptions on the class of distributions of the source and target." In fact, in P3 the class of pairs of distributions whose transfer distance is within a fixed number." is an assumption on distribution that is not trivial.
> >
> > 4. Point 4 claims, "First we would like to emphasize that our result is not asymptotic and holds for any number of samples."
> > In Theorem 1 and Theorem 2, the lower bounds contain c. When $n_S$ and $N_T$ are finite (not asymptotic), the value of constant c matters a lot in the bound. However, the value of c is not given in the paper. This may lead to a loose bound. In addition, the value of c is also needed in the experiment section to calculate the theoretical lower bounds with the value c.

---

> > > ### Author Response · Authors · 2021-11-29
> > > **Post Rebuttal**
> > >
> > > Thanks for the helpful comments.
> > >
> > > 1. We think that there has been a confusion in the comment. For a given realization of transfer learning problem, source and target distributions- which could be any arbitrary distributions- determine the value of $\Delta$. This $\Delta$ could be any number between [0,1] depending on the distance of source and target. We do not require that this fixed $\Delta$ be small or large. This $\Delta$ captures the distance of source and target tasks, and the distance could be either large or small. When the source and target are similar, the $\Delta$ would be small and when the source and target are not similar, the $\Delta$ would be large. In fact, an important consequence of our results is characterizing the regimes at which the source is very useful, moderately useful, or not useful at all. We state again that we don’t have any assumptions on $\Delta$. It just captures the distance of source and target which could be any arbitrary number. Furthermore, as the experiments demonstrate, there are source and target tasks whose corresponding $\Delta$ is small and there are source and target tasks whose corresponding $\Delta$ is large.
> > >
> > > With respect to the second part of the comment, this is not an assumption rather a natural formulation of transfer learning capturing the real world. In real datasets, when the source is not similar to the target, source samples do not compensate for the target samples, because there is a discrepancy between source and target. In other words, one needs to have access to enough number of target samples even if there is an abundant number of unrelated source samples. Our formulation captures this phenomenon, and we have not made any assumptions on the distributions. If the source is very close to the target, then it is very useful and if it is far from the target then it would be less useful, and our result captures this natural phenomenon. We want to emphasize further that even if the source is very close to the target but not exactly the same, it is natural that one still needs some target samples and cannot fully learn the target by using only the source.
> > >
> > > 2. Our goal is to develop a theoretical result that applies to practical settings. In the practical settings and contemporary models (e.g. deep learning), the order of VC dimension of networks is hundreds and thousands, and the assumption VC-dimension>10 is a very mild assumption. Therefore, this benign assumption does not imply any restrictions on our result in practical settings.
> > >
> > > 3. $\Delta$ is a function of the distributions, not the other way around. In other words, source and target distributions could be any arbitrary one, and they determine the value of $\Delta$. That is, $\Delta$ is a function of the distributions which capture the distance between the distributions. If the source and target distributions are close this would imply a small $\Delta$, and if the source and target distributions are far from each other, this would imply a large $\Delta$. This does not mean we have posed any restrictions on the distributions.
> > >
> > > 4. The constant c is given in Remark 3 and we have used that in our experiments.

---

### Official Review · Reviewer_9V1Q · 2021-11-03

**Correctness:** 3
**Technical Novelty And Significance:** 3
**Empirical Novelty And Significance:** 3
**Recommendation:** 6
**Confidence:** 4

**Main Review:**

The paper deals with a very interesting problem that is of fundamental importance: understanding the usefulness of transfer learning and its limitations given the relatedness between source and target domains.
Based on binary classification problems, the authors derive a lower bound for the generalization error that can be possibly achieved by any transfer learning algorithm as a function of source/target sample size.

Analysis based on real datasets for tasks like action recognition and image classification demonstrates how the derived bounds can be used to obtain interesting insights regarding transfer learning in practical scenarios.
Furthermore, following their theoretical derivations, the authors make a large number of remarks, which provide interesting and useful insights regarding transfer learning and enrich the discussions.

As the full knowledge of the source and target data distributions are not required in the analysis and as only minimal assumptions need to be made, the lower bound and the analysis presented in the paper may be relatively easily applicable to various transfer learning problems in a practical setting.


MAJOR CONCERNS

However, it is important to note that there exist relevant studies on binary classification in the context of transfer learning, where fundamental limits and efficacy of transfer learning have been discussed in depth.

Examples of such studies include:

1. Karbalayghareh, Alireza, Xiaoning Qian, and Edward R. Dougherty. "Optimal Bayesian transfer learning." IEEE Transactions on Signal Processing 66.14 (2018): 3724-3739.

2. Karbalayghareh, Alireza, Xiaoning Qian, and Edward R. Dougherty. "Optimal Bayesian transfer regression." IEEE Signal Processing Letters 25.11 (2018): 1655-1659.

3. Karbalayghareh, Alireza, Xiaoning Qian, and Edward Russell Dougherty. "Optimal bayesian transfer learning for count data." IEEE/ACM transactions on computational biology and bioinformatics (2019).

For example, in paper (1) shown above, its authors investigated the problem of optimal classification using both source domain and target domain data.
Uncertainty regarding the feature-label distributions was considered in the study, where a Bayesian scheme was used to derive the optimal Bayesian classifier that guarantees the best-expected classification performance.
This work investigated the effect of relatedness between domains as well as the complexity of the classification problem at hand to examine the benefits and limitations of transfer learning.

While the authors present a brief review of some prior work relevant to this current paper, it is narrowly focused on a few papers, failing to present the main contributions of the current study in light of relevant advances in the field, including the papers shown above.
Considering that there exists prior work that addresses (at least to some extent) questions such as "what is the best achievable accuracy via transfer learning for binary classification" and "how does this accuracy depend on the source/target domain sample size and the similarity between the domains", review of the relevant literature and presenting the proposed error bound and the analysis results in a proper context would be critically important.

Another concern about the current paper is that the experimental results based on the real datasets are somewhat limited to clearly demonstrate the benefits of the novel lower bound derived in the paper as well as the insights/remarks presented therein.
Additional examples would strengthen the work, and the authors may want to consider synthetic examples as well where the underlying feature-label distributions are known (with potential uncertainties).
For example, since the classification error of the optimal Bayesian transfer learning (OBTL) classifier can be estimated, it would be very interesting to consider such an example to test the tightness of the bound presented in this paper.


MINOR COMMENTS:

The current manuscript includes a large number of typos and grammatical errors.
Please carefully proofread the manuscript and correct the errors.

The authors may also want to show the error bounds as a function of source sample size - for different levels of relatedness/similarities between the source/target domains/tasks.
Currently, only the bounds as a function of target sample size are shown.




**Summary Of The Paper:**

In this paper, the authors aim to answer some of the fundamental questions regarding transfer learning.
Especially, the authors consider the problem of binary classification, for which they derive a novel lower bound on the generalization error achievable by any transfer learning algorithm as a function of source sample size and target sample size.
The presented results may improve our understanding of the utility and the limitations of transfer learning based on the relatedness between the source and target domains under consideration.
The authors show that the proposed method can be practically used in real applications, as it does not require full knowledge of the source and target domain distributions and makes minimal assumptions.
This paper considers real datasets for action recognition and image classification tasks for demonstration, based on which they also evaluate the sharpness of the derived bounds.



**Summary Of The Review:**

This paper investigates an interesting and important problem: namely, the fundamental limits of transfer learning in the context of binary classification.
The authors derive a novel lower bound on the generalization error, based on which they present useful insights regarding the efficacy and limitations of transfer learning, especially as a function of source/target sample size and the relatedness between the domains/tasks.
As the presented results only make minimal assumptions and do not require full knowledge of the source/target domain data distributions, the results in this current work may be relatively easily applied to various real-world transfer learning problems (for binary classification).
However, the current study overlooked closely relevant prior works on optimal transfer learning for classification (and other tasks), where relevant discussions on the efficacy and limitations of transfer learning have been made.
Furthermore, additional experimental results (e.g. based on synthetic examples) would be needed to clearly demonstrate the usefulness/accuracy of the derived lower bound and to further validate the insights presented in the paper.

---

> ### Author Response · Authors · 2021-11-22
> **Response to reviewer 9V1Q**
>
> Thanks for the comment. We have revised some of the typos and grammatical errors and will be further proof reading/revising these before the camera ready deadline. We also added a paragraph regarding the mentioned papers.
>
> Moreover, we also added some additional experiments regarding "error bounds as a function of source sample size" in the appendix of the paper.

---

### Official Review · Reviewer_WXNa · 2021-11-07

**Correctness:** 3
**Technical Novelty And Significance:** 2
**Empirical Novelty And Significance:** 1
**Recommendation:** 5
**Confidence:** 3

**Details Of Ethics Concerns:**

This reviewer does not have ethical concerns.

**Main Review:**

[Since it is an emergency review, I checked the proof sketch. The detailed derivations are not examined.]

Pros:
- This paper proposed a novel min-max learning bound under the proper transfer distance $\rho(P, Q)<1$. The conclusion is interesting and insightful.
- The proof is technically sound and recovers the conventional PAC-learning.

Cons:
- The proof seems straightly adopting the information-theoretical lower bound technique, where the key component (transfer distance) is generally **not** adopted in practice.
- The lower bound in multi-source is quite trivially extended.
- The concerns on empirical validation.

--------------------------------------------------------------------------------
Detailed Reviews:

- About the proof and significance.

The high-level proof idea is directly inspired by the conventional information-theoretical lower bound. The key difference lies in the introduction of transfer distance: the prediction risk gap (between the optimal source and target predictor) in the target domain. Although it is an interesting notion, as far as I know, almost no modern (in deep learning regime) transfer learning approach has indeed adopted this concept.

Besides, I have concerns about the paper scope. Since the papers in ICLR should generally involve the theory/practice w.r.t. the representation learning. The main contribution seems to have no direct relation to representation learning...

- About the multi-source scenarios

The multi-source scenario is just a trivial extension of the single source (by considering the $P_i-Q$ pairs). This can be fundamentally problematic (eq 4.2). In the multi-source scenarios, the $\Delta$ is not necessarily small. There can exist some poor sources, which is quite natural. However, the proof just simply assumes the small transfer distance, which is unreasonable. Thus the learnability is related to the proposed multi-source algorithm. The author is encouraged to check papers [1,2] for a better understanding. Paper [1,2] also involves the lower bound in the multi-source setting.

- About the experiments. The whole experiment is unclear

  1.  it is unclear how the lower bound is indeed justified in the experiments.
  2. The transfer risk is generally un-estimable since it is related to the ground-truth distribution. If we use observed samples to estimate, this can be quite problematic since the gap between the expected and observed terms is not proven.
  3. The same problem is shown in Fig 1(2), Fig 4.  Why weighted erm? What is the motivation for introducing weighted ERM? Multiple papers have adopted the weighted ERM with a clear upper bound. I could not understand the whole rationale in this setting.

- Other comments

   The transfer distance can be negative with a highly noisy target and clean source distribution. The author is encouraged to provide a discussion.

Reference: [1] A theory of multiple-source adaptation with limited target labeled data. Aistats 2021
[2] On the Sample Complexity of Adversarial Multi-Source PAC Learning. ICML 2020











**Summary Of The Paper:**

[Disclosure: This is an emergency review. ]

This paper demonstrates the min-max lower bound of transfer learning. (More precisely, transfer learning under the same binary label space $Y=(-1,+1)$). Eq (4.1) points out the key theoretical results. Then the bound is extended into the multi-source scenarios. The empirical validations are provided.

======Update after rolling discussion

I would like to appreciate the author for their detailed responses. I have read the whole reviews and responses. I decided to maintain my current rating, due to the concerns on theoretical and empirical significance.

**Summary Of The Review:**

This paper provided a lower bound in the binary domain adaptation classification problem, the conclusion is interesting and technically sound. However, this reviewer has concerns of significance and empirical validations. Based on this, I currently recommend a weakly negative score.

---

> ### Author Response · Authors · 2021-11-22
> **Response to Reviewer WXNa**
>
> We thank the reviewer for a thorough reading of the manuscript and helpful comments.
>
> - About the proof and significance
>
> From a practical point of view, practitioners try to come up with better and better algorithms for transfer learning, meaning that they try to achieve a better performance with limited number of target samples. The fundamental question in this context is to what extent source can be useful. In other words, if we see a saturation in practice is it because no more information from the source can be transferred or is it because of the fact that algorithms are not good enough.
>
> Our contribution is to come up with a notion of transfer distance and a lower bound that says no matter what algorithm one uses there is no way to improve the performance beyond the bound. This can be rather useful to practitioners as by assessing the gap between their scheme and the lower bound, they can understand potential algorithmic improvements that maybe possible with more clever schemes. This paper proves the first lower bound for transfer learning with minimal assumptions which can thus have considerable impact in terms of identifying deficiencies in algorithmic performance, guiding the development of new practical schemes.
>
> About the second part of the comment, one can view representation learning through the lens of transfer learning, because in transfer learning the goal is to learn a representation from the source that can be useful in the target. In that sense our result provides some guidelines what general representations from the source can be useful in the target. There is a long track record of transfer learning papers in ICLR so we think this paper is well suited to this venue.
>
> - About the multi-source scenarios
>
> The reviewer mentions that "In the multi-source scenarios, the $\Delta$ is not necessarily small". We agree with this statement and actually we did not assume small transfer distance in the multi-source case. In Theorem 2, every source can have an arbitrary transfer distance ($\Delta_i$) to the target which can be either small or large. Indeed, source number $i$ has transfer distance $\Delta_i$ which is different from other distances , and it  could also be any number in [0,1]. Hence the sentence "However, the proof just simply assumes the small transfer distance, which is unreasonable" is not true as we did not make any assumptions on the smallness of the transfer distances in Theorem 2.
>
>  Moreover, this theorem is not a trivial extension of Theorem 1, because in Theorem 1 we only have one transfer distance but in Theorem 2 we have multiple different transfer distances. Furthermore, In section 7.2, one can see that the construction of probability distributions for the packing sets is different from those considered in section 7.1, and by simply using Theorem 1 one cannot derive Theorem 2.
>
> As for the suggested papers by the reviewer, it is not clear to us how [2] is related to transfer learning directly. [2] studies a PAC learning setting where there are data from multiple sources generated by one single distribution $\mathcal{D}$, some of which are corrupted by an adversary. Furthermore, in [2] the distribution of the original training data and test data is the same which is substantially different from what is considered in our transfer learning problem.
>
>  [1] is a more related paper to our work and it proposes a new family of algorithms for multi-source domain adaptation problem. We have already cited several papers related to this line of work but we added this paper together with a discussion to the revised version. Thanks for pointing us to this paper.
>
> - About the experiments
>
> 1- Our main goal has been to derive a theoretical lower bound on the generalization error based on the available number of source and target samples. To empirically investigate the result, we estimated the parameters appearing in the lower bound using real datasets and we chose weighted ERM as an algorithm for deriving an upper bound for the target generalization error. The gap between the performance of upper bound and the lower bound is an indicator of how well the algorithms are performing.
>
> 2- In remark 7, we have explained that "one can estimate $Q[h_{S}^*(x_T)\neq y_T]$ rather accurately using a simple empirical average with a few target test data as well-known concentration of bounded functions imply that this empirical average is well concentrated around $Q[h^{*}_{S}(x_T)\neq y_T]$."
>
> 3- In order to investigate the sharpness of the lower bound, it suffices to use a specific algorithm whose error is close to the lower bound. So we just picked the weighted ERM as a baseline algorithm and then we observed that the error obtained by this algorithm is close to the lower bound. From this one can conclude that the lower bound is somewhat sharp.
>
> [1]On the Sample Complexity of Adversarial Multi-Source PAC Learning,[2] A Theory of Multiple-Source Adaptation with Limited Target Labeled Data.

---

> > ### Author Response · Authors · 2021-11-22
> > **Additional comments**
> >
> > - Other comments:
> >
> > Based on the transfer distance defined in this paper, when the the distance is 1 it means that source is very different from the target and the source data is not helpful in reducing the target error. Note that the lower bound gives the smallest achievable error considering both using and ignoring the source data. So when the distance is high one can just ignore the source data in order not to make the performance of the target worse. In other words, the concept of negative distance suggested by the reviewer is incorporated in our defined transfer distance.

---

> ### Comment · Reviewer_WXNa · 2021-11-23
> **Post Rebuttal**
>
> Dear author,
>
> I appreciate your detailed response. Below are my additional responses.
>
> 1. I know the gap between theory and practice. In the first question, I just point out the weakness of the proposed theory, i.e the empirical challenge of the transfer distance.
>
> 2. About paper scope. I still think the current paper may not be exactly suitable for ICLR. (I will discuss this later with other reviewers and AC). Although the lower bound holds for all the learning algorithms, it does not show the specific theoretical results for representation learning. Besides, the assumptions such as VC dim, 0-1 loss generally do not satisfy the modern representation scenarios, this paper may be suitable for more theoretical oriented conferences such as COLT, ICML, AIstats.
>
>     Please take note that I am not saying the contribution is not interesting, I personally feel the current version may be more suitable for the theoretical community to check the significance of the proof technique. As far as I understand, the proof technique is quite standard via information-theoretical lower bound.  But I could not judge the actual novelty since I am not the expert in min-max lower bound.
>
> 3. I still feel the significance in the multi-source is insufficient, given the related work of [1], which has much stronger theoretical and empirical results.
>
> 4. Why weighted ERM is closed to lower bound in **representation learning**? Please note the experiments are conducted in deep learning, why is it close? This is a really important question in addressing the scope for ICLR; if there is no formal theoretical understanding in the representation, this claim could be vacuous.
>
> 5. > well-known concentration of bounded functions imply that this empirical average is well concentrated around...
>
>      How well it is concentrated in the **representation learning**? Is it a vacuous result?
>
> Overall, it is highly expected to specify the theoretical contributions to meet the scope of representation learning. (such as non-vacuous concentration bound in representation learning)

---

> > ### Author Response · Authors · 2021-11-24
> > **Response to reviewer WXNa (Post Rebuttal)**
> >
> > We thank the reviewer for the helpful comments.
> >
> > In transfer learning the goal is to learn a good representation of the target domain using a related source data in order to achieve a small target generalization error. Some works in the transfer learning literature propose algorithms on how to learn the representation of the target domain, and by providing some upper bounds they guarantee that by using the proposed algorithms the error would not exceed a threshold.  On the other hand, some works in this literature derives impossibility results, meaning that by using any algorithms one can never get an accuracy better than a threshold by having access to a limited number of source and target samples. Our paper lies in the latter category and of course these two lines of works synergistically affect each other. Because if one does not have access to a lower bound on the generalization error, how can they make sure that the obtained representation is the best one and no one can produce a better representation?
> >
> > That said the “representation learning” in the name of the conference is meant to be interpreted broadly and it is quite clear that a paper on the fundamental limits of transfer learning is well-suited to ICLR.
> >
> > Through the remainder of the questions the reviewer is referring to a vague notion of **Representation Learning** which is not the subject of our paper. Without further clarification about what the reviewer means by “representation learning” in these questions, given that the paper is about transfer learning, is difficult to respond to. Would appreciate it if you can please clarify what you mean.
> >
> > With respect to the mentioned paper by the reviewer, we also agree with the reviewer that it has a significant contribution in the multi-source scenario. However, the mentioned paper’s contribution is different from ours in different ways. First, the focus of the paper is on proposing new algorithms for domain adaptation with some guarantees on the generalization error by proving upper bounds as they have explicitly stated in the abstract: “We show that a new family of algorithms based on model selection ideas benefits from very favorable guarantees “. In contrast, we propose lower bounds for any transfer learning algorithms in the multi-source scenario in the most general setting with minimal assumptions. Second, [1] assumes that the target distribution is close to some convex combination of sources as they have explicitly stated in the paper “we assume that the target distribution is close to some convex combination of sources in the discrepancy measure “. We however do not make assumptions about the relationship between source and target, and our bound is applicable to any setting. Finally, although they derive a lower bound working for any algorithms, they do not fix the hypothesis class for deriving the lower bound which is weaker than the setting in which one first fixes an arbitrary hypothesis class then derives a lower bound. Because in deriving a lower bound one considers the worst-case scenario and when the hypothesis class is not fixed the adversary has more freedom of choice to construct a worst-case scenario. But we first fix the hypothesis class and then derive the lower bound. We again state that [1] has a novel contribution in this literature but it is not directly comparable to our paper due to the aforementioned differences. We will add further discussion clarifying these differences.

---

### Decision · Program_Chairs · 2022-01-20

**Decision:**

Reject

**Comment:**

This paper shows minimax lower bounds on transfer learning for binary classification, in terms of a notion of transfer distance, defined in this paper. Experimental results try to show the validity of the proved minimax lower bounds.

All reviewers acknowledge that the lower bounds are worthy contributions; however, none of the reviewers felt strong enough to champion this work, due to that:
- the theoretical sharpness of the lower bounds are not discussed in detail (Reviewers TfQT, pmYF, and dGvD). Remark 6 only discusses the regime of a small amount of source data and a large transfer distance, which is fairly limited.
- it is unclear to what extent the experiments validates the theory (Reviewer WXNa). Note that for a minimax lower bound, for any algorithm, there is some corresponding "worst-case" datasets such that the algorithm does not do well; it is unclear if the datasets considered here are worst-case at all.
- the lower bound techniques are fairly standard.
- the comparisons between the lower bounds in this work and prior lower bounds (e.g. those in [1,2]) need to be discussed more thoroughly.

We encourage the authors to take into account the reviewers' feedback and revise the paper.

[1] Hanneke and Kpotufe. On the value of target data in transfer learning. NeurIPS 2019.
[2] ​​Mansour, Mohri, Ro, Suresh, and Wu. A theory of multiple source adaptation with limited target labeled data. AISTATS 2021.